# The Importance of the 5Cs of Positive Youth Development to Depressive Symptoms: A Cross-Sectional Study with University Students from Peru and Spain

**DOI:** 10.3390/bs13030280

**Published:** 2023-03-22

**Authors:** Denisse Manrique-Millones, Diego Gómez-Baya, Nora Wiium

**Affiliations:** 1Facultad de Ciencias de la Salud, Carrera de Psicología, Universidad Científica del Sur, Lima 15067, Peru; dmanrique@cientifica.edu.pe; 2Department of Social, Developmental and Educational Psychology, Universidad de Huelva, 21007 Huelva, Spain; diego.gomez@dpee.uhu.es; 3Department of Psychosocial Science, Faculty of Psychology, University of Bergen, 5020 Bergen, Norway

**Keywords:** depressive symptoms, Positive Youth Development, 5Cs, cross-national, Spain, Peru

## Abstract

Background: Prior research has documented the protective role of the 5Cs of Positive Youth Development (PYD) on adjustment problems, such as depressive symptoms. Nonetheless, more research is needed, especially in non-US contexts. The main objective of the present study was to assess associations between the 5Cs and depressive symptoms in Peru and Spain, considering gender differences across contexts. Methods: Cross-sectional data was collected from undergraduate students from Peru [*n* = 250] and Spain [*n* = 1044]. Results: The results revealed significant negative associations of Competence, Confidence, Character and Connection with depressive symptoms, while Caring was positively and significantly related to depressive symptoms in both samples. Regarding gender differences, female undergraduates in both samples reported high levels of Caring, while Competence was predominant among males compared to females in both countries. Likewise, higher scores in Competence and Confidence were registered among Peruvian male undergraduates compared to Spanish students, while Caring and Character were more prevalent in Spanish female undergraduates compared to Peruvian students. Conclusions: These findings confirm the importance of targeting the 5Cs of PYD alongside the role of gender and country context in intervention programs, put together to address the mental health of students in Peru and Spain.

## 1. Introduction

The 5Cs model of Positive Youth Development (PYD) is a widely accepted framework which proposes five psychological, behavioral and social qualities that indicate youth thriving. The 5Cs are Competence, Confidence, Character, Connection, and Caring [1]. Confidence reflects a positive sense of self-worth, mastery, future, positive identity and self-efficacy. Competence is a view of one’s capabilities with respect to a given domain or vocation. Character indicates respect for rules, morality and integrity. Caring is the ability to feel sympathy and empathy towards others. Finally, Connection concerns bonding with others and forming meaningful relationships with family, peers and communities [1]. Lerner proposes that when these 5Cs are present in a young person, a sixth C, contribution, emerges, where the young person can constructively contribute to self, family, community and even civil society (Lerner, 2004).

Anchored in the relational developmental systems theory, the PYD theoretical perspective suggests that the 5Cs is a function of the mutually influential relations that exist between young people and their context [2]. Accordingly, young people will thrive when their strengths are aligned with the opportunities and resources in youth contexts, such as families, schools and communities [2]. PYD also suggests that when youth are thriving, they are more likely to contribute to their environment, and they are also less likely to report adjustment problems, such as depression, delinquency and substance use [3].

Earlier studies evidenced that higher PYD scores were typically linked to fewer risk/adjustment problems, which reflect both internalizing problems such as depression and anxiety and externalizing problems such as bullying, substance abuse and delinquency [1,4]. To extend the generalizability of the PYD perspective outside the US context, we examine the influence of each component of the 5Cs of PYD on mental health in university students in Peru and Spain.

### 1.1. Depressive Symptoms and the Role of Positive Youth Development

Earlier studies have shown a sharp rise in depression rates at the beginning of adolescence that continue throughout the adolescence period [5,6]. Examining factors that may be helpful to safeguard youth mental health is essential given the functional impairment, increased risk of suicide and disturbed transition to adulthood that are related to depression [7].

In line with the World Health Organization [8], depression is a major cause of disability. The situation worsened during the COVID-19 pandemic. A recent scientific report published by the World Health Organization [9] indicated that during the first year of the pandemic, the prevalence of anxiety and depression increased dramatically across the globe by 25%. In this scenario, it is extremely important to explore factors that can contribute to protecting against mental health problems. Factors such as the 5Cs of PYD can be considered due to their protective effects against mental health problems, as well as their associations with well-being and flourishing in young people [10,11].

In the PYD literature, a negative association between four of the 5Cs [i.e., Competence, Connection, Confidence and Character] and depressive symptoms [12,13] as well as a positive association between Caring and depressive symptoms [13,14] have been observed. Holsen et al. [13] assessed the 5Cs of PYD and their association with maladaptive developmental outcomes, such as depressive symptoms among Norwegian students, and observed negative associations of the PYD indicators with depressive symptoms, as expected, with the exception of Caring which was positively associated with depressive symptoms. More recently, Gomez-Baya et al. [15] conducted a study with Spanish and Croatian students, which mirrored the aforementioned results in the sense that negative associations were found between Competence, Confidence, Connection and Character and depression, while the association with Caring was positive. 

In line with Geldhof et al. [14], the positive association between Caring and adjustment problems (i.e., depressive symptoms) can be explained by emotional sensitivity or a tendency for excessive concern about the feelings and thoughts of others. Accordingly, some young people who exhibit high levels of compassion and empathy, as captured in Caring, may also be more likely to exhibit higher levels of anxiety and depressive symptoms [13].

### 1.2. Gender Differences in Positive Youth Development and Depressive Symptoms

Earlier findings on gender differences in the 5Cs of PYD are diverse. According to Lerner et al. [16], females scored higher compared to males on an overall score of the 5Cs. Similarly, Zimmerman et al. [17] noted that females were more likely to score higher than males on an overall PYD factor. Notwithstanding, subtle disparities are seen when the 5Cs are examined separately. For instance, in a sample of Ghanaian adolescents, males scored higher in Competence while females reported more Caring [18]. Likewise, in a study involving Portuguese adolescents, males reported higher levels of Competence, Confidence and Connection compared to females [19]. These results suggest that the 5Cs of PYD may differ across gender.

Most of the research has explored the potential gender difference in samples of early adolescents [16,20], but less research has been conducted on university students. Gómez- Baya et al. [21,22] studied gender differences in a university student sample and discovered a similar pattern; males scored higher in Confidence and Competence, while females reported higher scores on Connection, Caring and Character compared to males.

It is worth noting that most prior research regarding gender differences in PYD has been done in the United States (e.g., [23,24]) with a recent growth in other contexts such as Europe (e.g., Norway; [25]; Ireland; [20]; Portugal; [19]) and Africa (e.g., Ghana; [18]). While a few PYD studies have also been done in the Latin American context (e.g., [26,27,28]), more research is needed in understudied samples to better understand gender differences in PYD indicators.

Research carried out concerning gender differences in depression indicates that depression is two to three times more prevalent in females than in males, predominantly at puberty [29,30]. Several possible explanations have been proposed, the most predominant being the gender roles hypothesis, social factors and biological factors. According to the gender roles hypothesis, female roles appear to be more restricted, conflict with societal responsibilities, and are usually undervalued [31]. Based on the “social factors” hypothesis, females are either more vulnerable to stressors associated with major life events or are differentially exposed to them [32,33]. Biological factors suggest that post-pubertal females are more inclined to exhibit limbic system hyperactivation in response to harmful stressors [34]. In the present study, we infer how these hypotheses on gender differences are reflected in the responses of our university students in Peru and Spain. 

### 1.3. Country Context: Peru and Spain 

Peru is located in western South America and is considered a region with high rates of violence [35] as a result of social inequity, affecting young people more tenaciously [36]. In Peru, young people [15–29 years] represent 26.9% of the total population. The OECD [37] estimates that two out of five young Peruvians are currently in a disadvantaged situation in terms of well-being. Indeed, depression is one of the predominant mental health issues among teenagers and young Peruvian adults [38].

Spain is in southwestern Europe and is considered the second largest country in the European Union. Despite having one of the largest economies in the eurozone [39], the youth unemployment incidence is high (i.e., 30.7%, [40]). Young people who remain inactive, neither studying nor working, increased during the pandemic from 19.2% to 22% [41].

It is important to note that in each country, the current pandemic may have had an impact to some extent on the population’s mental health due to the COVID-19 restrictions. Some studies have underscored the negative consequences of confinement on the mental health of Spanish youth [42,43]. In a longitudinal study by Ayuso-Mateos et al. [44], results indicated that younger individuals were at an increased risk of developing depression during the confinement. 

Regarding the Peruvian setting, studies showed that young adult students had a higher prevalence of mental health issues compared to other age groups [45]. In a large-scale study, the prevalence of depressive symptoms among Peruvian adults was five times higher than it was previously reported at a national level in 2018. Moreover, younger adults were more at risk to experience depressive symptoms compared to other age groups [46].

Given the situation experienced by both countries, the implementation of an intervention program that applies a strength-based perspective such as the 5Cs of PYD is promising if these thriving indicators are found to be protective against depressive symptoms.

### 1.4. Justification and Objectives of the Present Study

Even though some studies have been done on the association between thriving indicators and depressive symptoms with European samples (e.g., Croatia and Spain), more research such as the present study is needed in other cultural contexts and, especially, comparative ones. Moreover, if the protective role of the 5Cs is confirmed, future intervention that promotes these thriving indicators would help young people deal with their mental health as well in our Peruvian and Spanish contexts. Furthermore, although Peru and Spain are both Spanish-speaking countries, the cultural differences between them could reflect in the findings regarding the role of thriving indicators on depressive symptoms, a topic that warrants more attention and which ultimately can help tailor interventions to meet specific needs.

The present study aims are

(a)to analyze the association between the 5Cs and depressive symptoms among university students in Peru and Spain;(b)to examine gender differences in the 5Cs and depressive symptoms across countries;(c)and to study the indirect effects of the 5Cs on the country–depressive symptoms association and the gender–depressive symptoms association (Figure 1).

Based on previous research on internalizing behaviors, such as depression and anxiety, we expect to find a negative association between Character, Competence, Connection and Confidence and depressive symptoms [15,47] and a positive association between Caring and depressive symptoms [13,14]. 

In addition, we hypothesize that males will score higher in Competence, Confidence and Connection compared to females, while females will score higher in Caring and Character compared to males. Regarding depressive symptoms, based on earlier research, females would have a higher prevalence of depressive symptoms than males in both countries. Finally, we expect a mediation role of the 5Cs on the country–depressive symptoms association as well as the gender–depressive symptoms association.

## 2. Materials and Methods

### 2.1. Participants and Procedure

University students from Peru and Spain participated in the present study. As inclusion criteria, young people aged 18 to 29 years were selected to participate in the study.

The data involved 250 undergraduates from Peru (*M_age_* = 20.49; *SD* = 3.51), 60% females, and a sample of 1044 undergraduates from Spain (*M_age_* = 20.47; *SD* = 3.08), 75.5% females.

In Peru, cross-sectional data were collected from two private universities between March and June in 2021. First-year students registered in universities located in Metropolitan Lima were invited to participate in the study through convenience sampling. An online questionnaire was filled in by undergraduates who agreed to participate, taking about 25 to 30 min to complete the questionnaire. Prior to data collection, the students were informed of the study’s goals and anonymity. Informed consent was sought prior to their participation, which included assurances that participants could withdraw from the study at any time without incurring any penalties and that the outcomes would only be utilized for academic and research purposes. The study in Peru was approved by the Universidad San Martin de Porres ethical committee on 14 November 2019 (RD 3274, 14/11/2019). 

In Spain, a cross-sectional study was conducted from January to May in 2021. Participants were undergraduates who were enrolled in a total of 11 universities from different regions in Spain (University of Huelva, Loyola University [Campus of Seville and Cordoba], Complutense University of Madrid, University of Granada, University of Salamanca, University of La Laguna, University of Zaragoza, University of Santiago, Polytechnic University of Valencia, University of Valencia and University of Oviedo). These universities were selected through convenience sampling and controlled for geographical distribution (universities from the North, South, East and West of the peninsula, as well as island territory). However, university degrees were randomly selected in each university in Spain. The sample comprised undergraduates from all the academic years (1st year: 27.2%, 2nd year: 28.7; 3rd year: 23%, and 4th–6th year: 21.1%). No difference in depression was observed across academic year, *F*(3, 976) = 1.57, *p* = 0.195, nor in PYD, *F*(3, 981) = 0.93, *p* = 0.427.

After providing informed consent, an online survey was administered, having a duration of about 30 minutes, with measures for demographics, PYD, lifestyles and mental health. The study in Spain was part of the project “Positive Youth Development in Spanish Students” and received approval from the University of Huelva’s ethical committee on 10 January 2019 (UHU-1259711).

### 2.2. Measures

5Cs of Positive Youth Development—Short Form (PYD-SF; [14,22,27,47]). The adapted Spanish version of the PYD scale was used in this study. The questionnaire contains 34 items, which evaluate positive youth development indicators reflected in 5 dimensions: Competence (e.g., “I do very well in my classwork at school”), Confidence (e.g., “All in all, I am glad I am me”), Character (e.g., “I hardly ever do things I know I shouldn’t do”), Connection (e.g., “My friends care about me”) and Caring (e.g., “When I see another person who is hurt or upset, I feel sorry for them”). Responses ranged on a 5-point Likert-type scale, from 1 = strongly disagree to 5 = strongly agree, 1 = not important to 5 = extremely important, or 1 = not at all like me to 5 = very much like me. Good reliability scores were observed in the overall scale (α = 0.945) and in each of the 5Cs (Competence: α = 0.788; Confidence: α = 0.932; Connection: α = 0.874; Caring: α = 0.911; Character: α = 0.862) for the Peruvian sample, and acceptable coefficients were found in the overall scale (α = 0.876) and in four Cs (Competence: α = 0.725; Confidence: α = 0.771; Connection: α = 0.770; Caring: α = 0.822) with the exception of Character (α = 0.586), which was rather low, for the Spanish sample.

Depressive symptoms. The Patient Health Questionnaire 9 (PHQ-9; [48]) was used to examine depressive symptoms. This questionnaire was introduced with “How often have you been bothered by the following over the past 2 weeks?” and described 9 items (e.g., “Little interest or pleasure in doing things” and “Feeling down, depressed, or hopeless”). The response categories were on a 4-point Likert-type scale ranging from 0 = Not at all, to 3 = Nearly every day. To measure depressive symptoms, a sum score was calculated with a minimum score of 0 and maximum of 27. Internal consistency was very good in both samples: Peru (*α* = 0.907) and Spain (*α* = 0.853).

### 2.3. Data Analysis

Descriptive statistics were presented for the study variables (i.e., 5Cs of PYD, the overall mean score for PYD and depressive symptoms). Differences in gender, country and gender in each country were examined by conducting t-tests. Cohen’s d was reported for effect size. Then, Pearson correlations were estimated between the study variables in the total sample and by country. Next, linear regression analysis was carried out to examine depressive symptoms, with demographics (i.e., gender, age, country) and the 5Cs of PYD as independent variables. Moreover, the regression analysis was also conducted separately for each gender and country. For the findings, standardized coefficients and total R squared (i.e., variance explained) are reported. Finally, multiple partial mediation was tested to examine the indirect effects of each of the 5Cs (i.e., Competence, Confidence, Connection, Caring and Character) on the country–depressive symptoms association and gender–depressive symptoms association. Z values and confidence intervals are presented for total, indirect and direct effects and for residual covariances. In the analyses, results were considered significant at a level of *p* < 0.05. The analyses were carried out with JASP 0.16.1.0.

## 3. Results

### 3.1. Descriptive Statistics and Differences in Gender and Country

Table 1 presents the descriptive statistics (i.e., mean and standard deviation) of the 5Cs of PYD and the overall score for PYD and depressive symptoms for the total sample as well as by gender and by country. The moderate mean score was found for overall PYD (*M* = 3.70, *SD* = 0.47). Some differences were detected in the 5Cs: the highest mean scores were observed in Caring (*M* = 4.21, *SD* = 0.69) and Character (*M* = 3.96, *SD* = 0.51), while the lowest was found in Competence (*M* = 3.06, *SD* = 0.74). Concerning gender differences, the t-tests results indicated that males showed more Competence (*M* = 3.22, *SD* = 0.81) and Confidence (*M* = 3.77, *SD* = 0.75), while females reported higher Caring (*M* = 4.31, *SD* = 0.63) and Character (*M* = 3.99, *SD* = 0.51). No gender difference was found for Connection, the overall PYD score nor for the score on depressive symptoms. With regards to country differences, the Peruvian sample reported more Competence (*M* = 3.26, *SD* = 0.80) and Confidence (*M* = 3.86, *SD* = 0.88), while the Spanish sample reported more Caring (*M* = 4.32, *SD* = 0.58) and Character (*M* = 4.00, *SD* = 0.42). No country differences were found for depressive symptoms, Connection or the overall PYD score. Furthermore, for gender differences across countries, results in Table 2 indicate that in both Peruvian and Spanish samples, males scored higher in Competence, while females reported more Caring. Moreover, in the Spanish sample, females reported more Character than males. Furthermore, the interaction between gender and country was not significantly related to depressive symptoms, *F*(1, 1208) = 0.43, *p* = 0.514 nor to PYD, *F*(1, 1214) = 2.03, *p* = 0.154.

### 3.2. Bivariate Correlations between PYD and Depressive Symptoms

Table 3 presents Pearson correlations between PYD and depressive symptoms. The results showed a moderate negative correlation between overall PYD and depressive symptoms. Concerning the associations between the 5Cs and depressive symptoms, moderate negative correlations (*r* = −0.19–−0.49, *p* < 0.001) were found between depressive symptoms and Competence, Confidence, Connection and Character. However, a weak positive correlation was observed between depressive symptoms and Caring (*r* = 0.09, *p* < 0.01). Furthermore, the 5Cs showed positive interrelations. The strongest correlations were observed between Competence and Confidence and between Connection and Confidence. Table 4 presents the results for each country. A stronger association between overall PYD and depression was observed in the Spanish sample compared to the Peruvian sample. Furthermore, only in the Peruvian sample was Caring positively associated with Competence and Confidence.

### 3.3. Linear Regression Analyses 

Table 5 shows the results of the linear regression analyses of depressive symptoms, including demographics and the 5Cs as independent variables, with a repetition of the analyses for each gender and country. A check for any instances of autocorrelation in the regression analysis revealed a Durbin–Watson test for autocorrelation of 1.88, which indicated an acceptable value (normal range is 1.5 to 2.5). Furthermore, variance inflation factors (VIF) of the 5Cs were between 2.21 and 1.48, suggesting that there was no issue of multicollinearity (the generally accepted cut-off for VIF is 2.5).

In the total sample, country had a significant effect on depressive symptoms. Confidence and Connection were negatively associated with depressive symptoms, while Caring was positively associated with depressive symptoms. Furthermore, some gender differences were observed, where Confidence had the strongest negative association with depressive symptoms in the male sample, while Connection had the strongest negative association with depressive symptoms in the female sample. When comparing the influence on depressive symptoms by country, Confidence was negatively associated with depressive symptoms for Spain but not for Peru. Connection and Caring showed significant effects in both countries, whereas Connection had a negative association with depressive symptoms, while Caring showed a positive association. Furthermore, for gender differences by country, Connection was not significantly associated with depressive symptoms in Spanish males, while a strong negative association was found in Peruvian males. Moreover, a strong positive association with Caring was found in Peruvian females. Finally, concerning the explanatory power of the regression models, the results showed that, in Peru, gender, age and the 5Cs explained 18.6% of the variance in depressive symptoms, while in Spain, the explained variance was 37.8%. 

### 3.4. Multiple Partial Mediation Model

Figure 1 presents the multiple partial mediation model tested, in which the influences of gender and country on depressive symptoms were partially mediated by the 5Cs of PYD. Table 5 shows the results of the multiple partial mediation model. Concerning direct effects, only country had a significant association with depressive symptoms. With regards to indirect effects, Confidence and Caring were found to mediate the country–depressive symptoms association. Thus, country differences in Confidence and Caring partly explained country differences in depressive symptoms. The higher score in Confidence and the relatively lower score in Caring in the Peruvian sample (compared to the Spanish sample) may have provided some protection against depression, as high scores in Caring were found not to be protective. Moreover, Caring was found to mediate the influence of gender on depressive symptoms, so gender differences in Caring could partly explain the association between gender and depressive symptoms (Table 6). Specifically, the relatively lower score in Caring in the male sample may provide some protection against depressive symptoms compared to the female sample. Further analysis to compare the indirect effects of Confidence and Caring on the association between country and depressive symptoms indicated that Caring had greater effect (Z = 5.56, *p* < 0.001, 95% CI [−1.04, −0.50]) than Confidence (Z= −4.20, *p* < 0.001, 95% CI [−1.35, −0.49]). Finally, residual covariances between the 5Cs were found to be significant, so positive interrelations were detected among these dimensions of PYD.

## 4. Discussion

The main objective of the present study was to examine associations between the 5Cs of PYD and depressive symptoms in two countries: Peru and Spain. Our research question was threefold.

First, we aimed to analyze the relationships between the 5Cs and depressive symptoms in two different contexts. In line with previous research, our hypothesis that Connection, Character, Confidence and Competence would be negatively associated with depressive symptoms was confirmed, suggesting that higher levels of positive youth development indicators were associated with fewer depressive symptoms in Peruvian and Spanish university students. The results shed light on the important role of PYD indicators on depressive symptoms in Peru and Spain.

Consistent with how Confidence is operationalized in the present study as an internal sense of self-worth and self-efficacy, previous research had highlighted that low self-esteem is associated with depressive symptoms. Young people who scored low on internal sense of self-worth had a higher risk of developing symptoms of depression [49,50,51]. In a cross-sectional study among Vietnamese students, anxiety, depression and suicide ideation were associated with low self-esteem [52], indicating that poor mental health was linked to low self-esteem.

In the case of Connection, some studies have commented on its protective effects against depressive symptoms [53]. For instance, Choi et al. [54] stated that even for people who had a higher chance of developing depression due to a genetic predisposition or early-life stress, social connections had protective effects. It is therefore important to nurture and promote Confidence and Connection among young adults to prevent depressive symptoms. Corresponding to earlier findings (e.g., [55]), our results showed a positive although weak association between Caring and symptoms of depression. The finding may suggest that high levels of compassion reflected in Caring can generate hypersensitivity in university students, making them prone to develop depressive symptoms. We can speculate that this scenario was aggravated by the numerous infections and fatal cases brought about by the COVID-19 pandemic, especially in Spain, which was among the first countries to suffer from the pandemic.

Regarding the levels of the 5Cs in each country, Peruvian undergraduates appeared to report more Confidence and Competence compared to their European peers, while Spanish undergraduates appeared to display higher tendency towards Caring and Character. Interestingly, these two pairs of Cs are often found in tandem, with the Confidence–Competence pair known as the efficacious Cs and the Caring–Character pair known as the socio-emotional Cs [14]. High scores on self-reported Caring may imply emotional hypersensitivity, which may be a risk factor for depression. In intervention situations, adequate levels of the 5Cs should be the target as that can eventually lead to the development of the sixth C of Contribution, where young people can contribute to self and society [27,56].

Second, we analyzed gender differences in the 5Cs and depression symptoms across countries. Regarding gender differences in depressive symptoms in comparison to the general population’s prevalence before the pandemic, there has been, in both countries, a recorded increase in the incidence of anxiety and depressive symptoms during the COVID-19 pandemic. Studies have suggested that females are more at risk to develop depression and, in general, mental health problems compared to males [57]. Nonetheless, in the present study we did not find any significant gender difference in depressive symptoms. It is important to note that the mean score of depressive symptoms in our sample was 9.10, meaning that some depression was reported although not pathological levels. While the levels of depressive symptoms could have been affected by the pandemic restrictions, the non-pathological levels of depressive symptoms that were reported could account for the lack of gender differences. One could also argue that all students in our sample were equally exposed to COVID-19 restrictions, such as quarantine, isolation and obligatory use of face masks. 

In relation to gender differences associated with the 5Cs, the hypothesis that male students would report higher levels of Competence and Confidence while female students would report higher levels of Caring and Character was confirmed. Women are traditionally characterized by their social skills and found to be more expressive and sensitive on emotional levels, while displaying more agreeability, care towards personal relationships and empathy [58], all of which are values that align with Caring and Character. Conversely, men are more worried about achievements and results [59], which tend to reflect Competence and Confidence. This trend was consistent in Peru and Spain with males scoring higher in Competence and females reporting higher levels of Caring in both countries, while females also reported higher levels of Character in Spain.

Regarding indirect effects of the 5Cs, interestingly we found partial mediation of PYD indicators on the association of country with depressive symptoms, wherein Confidence and Caring in Peruvian undergraduates appeared to be protective against depressive symptoms. As it has been discussed previously, higher rates of self-efficacy constitute a shield against self-depreciating behaviors and, consequently, distress [50,51]. 

Despite the significant results in the regression analyses, caution must be taken when interpreting the findings, as the proportion of variance explained (mainly for the Peruvian sample) was somewhat low and might indicate that other factors could play a more important role in the prediction of depressive symptoms. 

### 4.1. Limitations

The present study has some limitations worth noting. First, the sample sizes of the two countries were uneven, with more participation from Spanish undergraduates. Future research studies should aim at having balanced samples. Second, the sampling technique in both countries was convenience, with Peru especially having specific characteristics which might not represent university students as a whole. Future research can use probabilistic sampling techniques to select a representative sample. Third, in line with the previous limitation, we have focused the research in the urban areas, which can hinder the generalization of the results. Data on several social background variables such as social inequality and other factors related to confinement, which could have been considered in the analysis, were also not collected. Moreover, the data were collected during the pandemic, which could have affected participants’ responses, especially those related to depressive symptoms. Future research should expand the sample into rural zones and actively examine the effect of social variables as well as the study variables and the different associations in post-pandemic samples. Fourth, the data collection was performed online; thus, while students were enrolled in universities located in their respective cities, their geographical location during the data collection could be elsewhere, which could impact their responses and, ultimately, our results. It is necessary that forthcoming research take this inconsistency into account. Fifth, the reliability coefficient for the PYD dimension Character in the Spanish sample was rather low; thus, results involving this variable should be interpreted with caution. Sixth, sociodemographic data were limited in the present study, and it is recommended that subsequent studies consider demographics related to the region of origin, current residence, or socioeconomic status. Finally, the cross-sectional nature of the data which focuses on a specific period of time prevents us from investigating trajectories of the variables and causality. It is recommended to conduct longitudinal studies in which trends and developmental effects of the 5Cs on depressive symptoms can be observed.

### 4.2. Implications for Research, Policy and Practice

Despite the limitations, this study has some important implications for research, policy and practice. 

One implication for research is the importance of considering PYD in terms of its dimensions (the 5Cs) and not merely as a global factor when observing its impact on youth development (in this case, university students). It is crucial to analyze the effect of each of the 5Cs with context awareness. Although the 5Cs act as adaptive regulators, sub-optimal levels can make them maladaptive, as in the case of Caring. Consequently, one should be wary of overstating the role of Caring in intervention set up to address depressive symptoms.

Furthermore, public policies must respond to the vulnerability of young Peruvians and Spanish youth, where PYD indicators can be nurtured in different youth contexts (e.g., neighborhood, family, educational environments). As has been evaluated in other settings (e.g., US; [60]), some community-based programs employing PYD can reduce the impact of risk factors [61]. Finally, although not at pathological levels, some depressive symptoms were reported by Peruvian and Spanish youth, suggesting that intervention programs at universities should have routine monitoring of students’ mental health as well as intervention strategies that can prevent the development of pathological levels of depression.

Concerning implication for practice, intervention strategies can be built on PYD indicators such as the 5Cs that were found to be negatively associated with depressive symptoms, considering some country variations. In this sense, it is important that researchers and practitioners promote not only high but optimal levels of the 5Cs (especially Caring) as well as work to create and execute culturally tailored PYD programs. 

## 5. Conclusions

This study contributes to the ongoing expansion of research on PYD, investigating associations between the 5Cs of PYD and depressive symptoms in vulnerable contexts, both highly affected by the current pandemic. First, Competence, Confidence, Connection and Character were negatively associated with depressive symptoms across samples, findings which can guide our focus in intervention or prevention programs. 

Second, Caring was found to be positively associated with symptoms of depression, meaning that high levels of Caring can be a risk factor for depressive symptoms among undergraduates in both Peru and Spain. Thus, one should take this into account when addressing these symptoms in university students. Third, in both countries, female students reported higher levels of Caring, which extend the PYD literature on gender differences. Fourth, Competence and Confidence were found to be stronger among Peruvian male students, while Caring and Character were more prevalent in Spanish female students, suggesting some country differences in the 5Cs. Fifth, results did not reveal any gender differences regarding depressive symptoms, which could be explained by the pandemic restrictions and consequences. Finally, we found partial mediation of PYD indicators on the association between country and depressive symptoms, where Confidence and Caring were the only significant mediators for Peruvian undergraduates. The current findings can help inform research, policy and programs that seek to enhance thriving and mental health in young people, especially in Peru and Spain.

## Figures and Tables

**Figure 1 behavsci-13-00280-f001:**
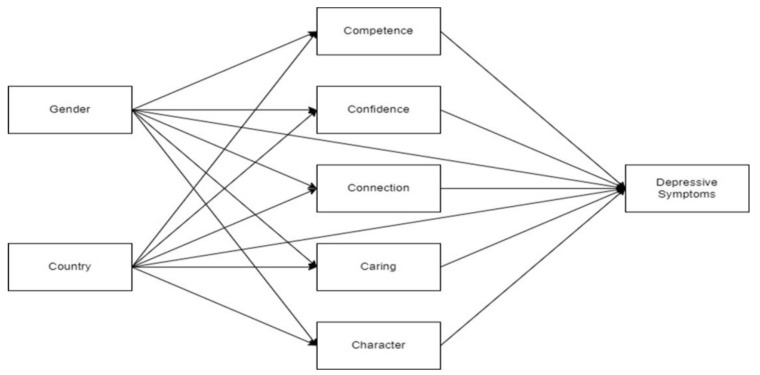
Proposed Multiple partial mediation model of the effects of gender and country on depressive symptoms through the 5Cs of PYD.

**Table 1 behavsci-13-00280-t001:** Gender and country differences in the 5Cs of PYD and depression symptoms, t-test analysis.

	Total M(SD)	Female M(SD)	Male M(SD)	Peru M(SD)	Spain M(SD)	Gender Differences	Country Differences
Competence	3.06(0.74)	3.00(0.70)	3.22(0.81)	3.26(0.80)	3.01(0.71)	t(1231) = 4.88, *p* < 0.001, d = 0.312	t(1231) = −5.00, *p* < 0.001, d = −0.354
Confidence	3.68(0.74)	3.65(0.74)	3.77(0.75)	3.86(0.88)	3.64(0.70)	t(1231) = 2.52, *p* = 0.012, d = 0.161	t(1231) = −4.28, *p* < 0.001, d = −0.303
Connection	3.60(0.65)	3.62(0.62)	3.56(0.70)	3.56(0.76)	3.61(0.61)	t(1216) = −1.46, *p* = 0.144, d = −0.094	t(1216) = 1.08, *p* = 0.279, d = 0.077
Caring	4.21(0.69)	4.31(0.63)	3.94(0.76)	3.77(0.87)	4.32(0.58)	t(1222) = −8.60, *p* < 0.001, d = −0.552	t(1222) = 11.88, *p* < 0.001, d = 0.842
Character	3.96(0.51)	3.99(0.51)	3.87(0.52)	3.79(0.75)	4.00(0.42)	t(1231) = −3.61, *p* < 0.001, d = −0.230	t(1231) = 5.87, *p* < 0.001, d = 0.416
Overall PYD	3.70(0.47)	3.71(0.46)	3.67(0.51)	3.65(0.63)	3.71(0.42)	t(1216) = −1.32, *p* = 0.186, d = −0.085	t(1216) = 1.92, *p* = 0.055, d = 0.136
Depressive symptoms	9.10(5.89)	9.29(5.84)	8.60(6.00)	9.67(7.05)	8.95(5.54)	t(1210) = −1.84, *p* = 0.067, d = −0.118	t(1210) = −1.71, *p* = 0.088, d = −0.121

**Table 2 behavsci-13-00280-t002:** Gender differences in the 5Cs of PYD and depressive symptoms among students in Peru and Spain, t-test analysis.

	Peru		Gender Differences in Peru	Spain		Gender Differences in Spain
	Female	Male		Female	Male	
Competence	3.15(0.75)	3.44(0.83)	t(248) = 2.89, *p* = 0.004, d = 0.373	2.96(0.68)	3.13(0.78)	t(981) = 3.17, *p* = 0.002, d = 0.236
Confidence	3.82(0.90)	3.93(0.86)	t(248) = 0.95, *p* = 0.341, d = 0.123	3.62(0.70)	3.70(0.68)	t(981) = 1.68, *p* = 0.093, d = 0.125
Connection	3.56(0.75)	3.56(0.79)	t(248) = −0.05, *p* = 0.963, d = −0.006	3.63(0.60)	3.56(0.66)	t(966) = −1.58, *p* = 0.115, d = −0.118
Caring	3.87(0.82)	3.63(0.91)	t(248) = −2.11, *p* = 0.036, d = −0.273	4.40(0.54)	4.07(0.64)	t(972) = −7.69, *p* < 0.001, d = −0.576
Character	3.77(0.81)	3.82(0.65)	t(248) = 0.51, *p* = 0.609, d = 0.066	4.04(0.41)	3.90(0.46)	t(981) = −4.47, *p* < 0.001, d= −0.333
Overall PYD	3.63(0.64)	3.68(0.62)	t(248) = 0.52, *p* = 0.603, d = 0.067	3.73(0.41)	3.67(0.45)	t(966) = −1.83, *p* = 0.067, d = −0.138
Depressive symptoms	10.16(7.28)	8.93(6.66)	t(248) = −1.35, *p* = 0.177, d = −0.175	9.11(5.48)	8.46(5.70)	t(960) = −1.58, *p* = 0.115, d = −0.119

**Table 3 behavsci-13-00280-t003:** Bivariate correlations between the 5 Cs of PYD and depressive symptoms for the total sample.

	1	2	3	4	5	6	7
1. Competence	-						
2. Confidence	0.64 ***	-					
3. Connection	0.49 ***	0.54 ***	-				
4. Caring	0.07 *	0.05	0.27 ***	-			
5. Character	0.32 ***	0.47 ***	0.45 ***	0.50 ***	-		
6. Overall PYD	0.74 ***	0.78 ***	0.77 ***	0.51 ***	0.74 ***	-	
7. Depressive symptoms	−0.34 ***	−0.49 ***	−0.39 ***	0.09 **	−0.19 ***	−0.39 ***	-

* *p* < 0.05, *** p* < 0.01, *** *p* < 0.001.

**Table 4 behavsci-13-00280-t004:** Bivariate correlations between the 5Cs of PYD and depressive symptoms by country.

	1	2	3	4	5	6	7
1. Competence	-	0.73 ***	0.49 ***	0.31 ***	0.55 ***	0.79 ***	−0.26 ***
2. Confidence	0.60 ***	-	0.54 ***	0.28 ***	0.64 ***	0.82 ***	−0.30 ***
3. Connection	0.50 ***	0.55 ***	-	0.43 ***	0.56 ***	0.77 ***	−0.30 ***
4. Caring	0.05	0.01	0.20 ***	-	0.58 ***	0.67 ***	0.13 *
5. Character	0.27 ***	0.43 ***	0.40 ***	0.42 ***	-	0.85 ***	−0.14 *
6. Overall PYD	0.75 ***	0.78 ***	0.78 ***	0.44 ***	0.67 ***	-	−0.22 ***
7. Depressive symptoms	−0.39 ***	−0.59 ***	−0.43 ***	0.10 **	−0.21 ***	−0.47 ***	-

Note. The results for the Spanish sample are presented below the diagonal, and the results for the Peruvian sample are shown above the diagonal. * *p <* 0.05, *** p* < 0.01, *** *p* < 0.001.

**Table 5 behavsci-13-00280-t005:** Linear regression analyses to explain depressive symptoms based on the 5Cs of PYD by country and gender.

		Total		Female		Male	
		F/R^2^	β	F/R^2^	β	F/R^2^	β
Total sample		69.42 ***/0.312		55.80 ***/0.305		23.59 ***/0.323	
	Gender		0.01				
	Age		−0.03		−0.04		−0.02
	Country		0.17 ***		0.18 ***		0.15 **
	Competence		−0.02		−0.04		0.05
	Confidence		−0.40 ***		−0.38 ***		−0.50 ***
	Connection		−0.24 ***		−0.26 ***		−0.18 **
	Caring		0.20 ***		0.18 ***		0.20 ***
	Character		0.04		0.06		0.01
Peru		9.11 ***/0.186		5.68 ***/0.158		5.78 ***/0.224	
	Gender		0.02				
	Age		−0.06		−0.10		0.02
	Competence		−0.08		−0.08		−0.05
	Confidence		−0.16		−0.13		−0.22
	Connection		−0.31 ***		−0.22 *		−0.46 ***
	Caring		0.33 ***		0.39 ***		0.29 *
	Character		−0.01		−0.14		0.14
Spain		83.92 ***/0.378		72.29 ***/0.371		30.09 ***/0.429	
	Gender		0.01				
	Age		−0.01		−0.02		0.01
	Competence		−0.02		−0.03		0.07
	Confidence		−0.48 ***		−0.45 ***		−0.65 ***
	Connection		−0.20 ***		−0.26 ***		−0.02
	Caring		0.13 ***		0.10 **		0.17 **
	Character		0.02		0.07		−0.07

Note. * *p* < 0.05, ** *p* < 0.01, *** *p* < 0.001.

**Table 6 behavsci-13-00280-t006:** Multiple partial mediation model of the effects of gender and country on depressive symptoms through the 5Cs of PYD, direct, indirect, total effects, and residual covariances.

					95% CI	
	Estimate	Std. Error	z-Value	*p*	Lower	Upper
Direct effects						
Country → PHQ9	2.41	0.37	6.46	<0.001	1.68	3.15
Gender → PHQ9	0.05	0.33	0.14	0.886	−0.60	0.69
Indirect effects						
Country → Competence → PHQ9	−0.04	0.06	−0.73	0.468	−0.16	0.07
Country → Confidence → PHQ9	−0.66	0.18	−3.75	<0.001	−1.01	−0.32
Country → Connection→ PHQ9	0.09	0.10	0.91	0.366	−0.10	0.28
Country → Caring→ PHQ9	−0.86	0.15	−5.70	<0.001	−1.16	−0.57
Country → Character → PHQ9	−0.09	0.08	−1.16	0.247	−0.23	0.06
Gender → Competence → PHQ9	0.04	0.05	0.73	0.469	−0.06	0.14
Gender → Confidence → PHQ9	0.29	0.15	1.92	0.055	−0.01	0.59
Gender → Connection→ PHQ9	−0.12	0.09	−1.29	0.199	−0.29	0.06
Gender → Caring→ PHQ9	0.52	0.11	4.94	<0.001	0.32	0.73
Gender → Character → PHQ9	0.04	0.04	1.10	0.274	−0.03	0.11
Total effects						
Country → PHQ9	0.85	0.42	2.02	0.044	0.02	1.67
Gender → PHQ9	0.83	0.38	2.17	0.030	0.08	1.57
Total indirect effects						
Country → PHQ9	−1.57	0.27	−5.82	<0.001	−2.09	−1.04
Gender → PHQ9	0.78	0.23	3.42	<0.001	0.33	1.23
Residual covariances						
Competence ↔ Confidence	0.34	0.02	18.74	<0.001	0.30	0.37
Competence ↔ Connection	0.24	0.02	15.84	<0.001	0.21	0.27
Confidence ↔ Connection	0.26	0.02	16.79	<0.001	0.23	0.29
Competence ↔ Caring	0.07	0.01	5.38	<0.001	0.05	0.10
Confidence ↔ Caring	0.05	0.01	3.64	<0.001	0.02	0.08
Connection ↔ Caring	0.11	0.01	9.10	<0.001	0.09	0.13
Competence ↔ Character	0.13	0.01	12.12	<0.001	0.11	0.16
Confidence ↔ Character	0.19	0.01	15.82	<0.001	0.16	0.21
Connection ↔ Character	0.15	0.01	14.29	<0.001	0.13	0.17
Caring ↔ Character	0.15	0.01	15.12	<0.001	0.13	0.17

Note. Delta method standard errors, normal theory confidence intervals, ML estimator.

## Data Availability

The data presented in this study are available on request from the corresponding author.

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
