# Peer review of "The Importance of the 5Cs of Positive Youth Development to Depressive Symptoms: A Cross-Sectional Study with University Students from Peru and Spain"

_behavsci, 2023, doi:10.3390/bs13030280_

Round 1
Reviewer 1 Report
Dear authors,
I am glad to contribute to the peer review of your manuscript. You have invested a lot of time and effort in carrying out this research and in writing this manuscript. I will now give you some advice and recommendations, based on my experience and expertise, to improve your manuscript. Please, let me know if you have any doubt about my comments.
Brief summary
Main goals: The present study aims to analyze the associations between the 5Cs of PYD and depression in the Latin American and European contexts, inferring potential cultural differences. The specific goals are:
(a) To analyze the association between the 5Cs and depression among university students in Peru and Spain.
(b) To examine gender differences in the 5Cs and depression symptoms across countries.
(c) And to study the indirect effects of the 5Cs on the country-depression association and the gender-depression association
Main contributions: Expanding the literature about PYD and its relation to depression.
Strengths: Large sample from two countries, a solid theoretical framework (5Cs model of PYD), proper analysis and discussion of data.
General comments
· Is the manuscript clear, relevant for the field and presented in a well-structured manner?
The manuscript is usually clear, but there is a need of reviewing language and writing style, especially in the introduction. This is not clear the relevance for the field.
· Are the cited references mostly recent publications (within the last 5 years) and relevant? Does it include an excessive number of self-citations?
There are several recent references from the last 5 years, and the number of self-citations is adequate.
· Is the manuscript scientifically sound and is the experimental design appropriate to test the hypothesis?
There could be a lack of validity in the PYD-SF questionnaire used, since the Spanish translation used has not been validated previously, and the original instrument was not validated for the ages of the participants of the present study.
· Are the manuscript’s results reproducible based on the details given in the methods section?
Yes
· Are the figures/tables/images/schemes appropriate? Do they properly show the data? Are they easy to interpret and understand? Is the data interpreted appropriately and consistently throughout the manuscript? Please include details regarding the statistical analysis or data acquired from specific databases.
Yes
· Are the conclusions consistent with the evidence and arguments presented?
Yes
· Please evaluate the ethics statements and data availability statements to ensure they are adequate.
Ethics statement and data availability statement are adequate.
Specific comments
1. Line 13: I believe you are referring to depression as a problematic behavior, not a problem behavior.
2. Please carefully check the writing of the abstract. There are some grammatical mistakes that should be taken care of. It is difficult to read, especially because of the absence of commas.
3. English should be revised throughout the manuscript, since there are several grammatical, verbal and spelling mistakes.
4. In the first paragraph of the introduction (lines 31-39), the five Cs of Lerner et al. are explained. However, there is no information about the sixth C, contribution, which is the main goal of this specific PYD framework. Contribution arises when the other five Cs are present, I think there should be at least a mention about it.
5. Line 40, you say PYD for referring to positive youth development. However, previously you do not mention anything about PYD meaning. Please, include this acronym between brackets the first time the term positive youth development is used in the main text.
6. Line 46, there should be stated “problematic behaviors” instead of “problem behaviours”. Please use the same style of English, either American or British (behavior/behaviour).
7. I am not really sure whether depression is a problematic behavior like violence, delinquency or drugs/alcohol use. Why do you consider it as a problematic behavior instead of an illness?
8. Please check and correct throughout the whole manuscript the term “problem behavior”.
9. Some connectors do not make sense with the information of the text (e.g. line 81).
10. L110: Prominent. Do you mean predominant?
11. Why do you believe this research is important for PYD literature and depression field of study? This reflection should be included in section 1.4 (L157). Reading the introduction and theoretical framework, I still do not clearly see why is this research important, what is the differential element that makes it useful for the field of study. There are plenty of references addressing PYD (and specifically the 5Cs model) and depression. What is different with your study?
12. In the participants section, is there any selection criteria for the sample? In university, there can be people from 18 to 65+ years, but PYD framework addresses only youth people.
13. Checking the validation of the instruments used, I see that the original questionnaire designed in the US (reference 14) was validated for grade 5 to grade 12 students (i.e. up to 18 years, what in Spanish would be “bachillerato”). However, the sample from this study is older than that. Has there been any validation on people up to 29 years? (I believe that is the age limit you used, since it is not clear in the manuscript and is only addressed in the introduction as young adults up to 29 years).
Moreover, when it comes to the Spanish translation of the instrument, I cannot see any validation in the reference 21 regarding the PYD-SF. This reference only shows a cross-sectional study from the 2nd author of the present manuscript in which a translation of PYD-SF has been carried out, but it has not been validated. I also miss the validation in terms of age, as I stated previously.
14. Character’s α value in Spanish sample is low to be considered acceptable.
15. In 3.1 section, the Connection results are missing in the first paragraph (L273-282). The other Cs are all addressed.
16. L343 – Change the square brackets for parentheses.
17. L416 – Please change “preoccupied with” for “worried about”
18. L420 – Indent in first line is missing
19. It could be interesting for your further research to address more than one theoretical model of PYD. There are several PYD frameworks that could help you to gain insight into the PYD-depression field, such as the Developmental Assets Profile (Benson, 1997), the Domains of Learning Experiences (Dworkin et al., 2003; Larson et al., 2006), or the more recent Critical PYD (Gonzalez et al., 2020).
In Spain, Oliva et al. (2010) designed an educational PYD model through teamwork among psychologists, psychiatrists and education professionals (Oliva, A., Ríos, M., Antolín, L., Parra, Á., Hernando, Á., & Pertegal, M.-Á. (2010). Más allá del déficit: construyendo un modelo de desarrollo positivo adolescente. Infancia y Aprendizaje, 33(2), 223-234. https://doi.org/10.1174/021037010791114562).
I think the work from Olenik (2019) could also help you to dig deeper and gain more insight about this topic (Olenik, C. (2019). The Evolution of Positive Youth Development as a Key International Development Approach. Global Social Welfare, 6(1), 5-15. https://doi.org/10.1007/s40609-018-0120-1)
20. L501 – please write correctly the Data Availability Statement
I hope my comments will help you to enhance your manuscript and your research. Best regards.
Author Response
We would like to express our gratitude to the Editor and the Reviewers for their comments and suggestions, which helped us to strengthen the paper. In order to show how we have addressed each comment/suggestion, and to which changes it led in the manuscript, we have broken the reviews into as many points as it contained. Each point has been numbered and addressed in turn. Reviewers’ comments are in bold. Corresponding changes implemented in the manuscript appear in blue for the convenience of the Editor and the Reviewers.
Reviewer 1
I am glad to contribute to the peer review of your manuscript. You have invested a lot of time and effort in carrying out this research and in writing this manuscript. I will now give you some advice and recommendations, based on my experience and expertise, to improve your manuscript. Please, let me know if you have any doubt about my comments.
General comments
- Is the manuscript clear, relevant for the field and presented in a well-structured manner?
The manuscript is usually clear, but there is a need of reviewing language and writing style, especially in the introduction. This is not clear the relevance for the field.
Reply: Thank you. We have now reviewed the language and writing style.
- Are the cited references mostly recent publications (within the last 5 years) and relevant? Does it include an excessive number of self-citations?
There are several recent references from the last 5 years, and the number of self-citations is adequate.
Reply: Thank you for your comment.
- Is the manuscript scientifically sound and is the experimental design appropriate to test the hypothesis?
There could be a lack of validity in the PYD-SF questionnaire used, since the Spanish translation used has not been validated previously, and the original instrument was not validated for the ages of the participants of the present study.
Reply: The psychometric properties of the 5Cs of PYD scale has been verified in American adolescent samples (Bowers et al., 2010; Geldhof et al., 2014; Phelps et al., 2009) as well as in emerging adults (Dvorsky et al., 2019). Additionally, its validity has been confirmed in many other contexts, for example, samples of young people in Europe, in Ireland (Conway, 2015), in Lithuania (Erentaite & RaižienÄ—, 2015), in Italy, Bulgaria, Rumania and Norway (Dimitrova et al., 2021) and also outside Europe (i.e., New Zealand, Fernandes et al., 2021). In the Spanish context the psychometric properties of the scale have been confirmed in previous studies (Gómez-Baya et al., 2019; Kozina et al., 2021), and also in South America, Perú and Colombia (Manrique-Millones, 2021). We have added these references in the manuscript.
Dvorsky, M. R., Kofler, M. J., Burns, G. L., Luebbe, A. M., Garner, A. A., Jarrett, M. J., et al. (2019). Factor structure and criterion validity of the Five Cs model of positive youth development in a multi-university sample of college students. Journal of Youth and Adolescence, 48(3), 537–553. https://doi.org/10.1007/s10964-018-0938-y
Gomez-Baya, D., Reis, M., and Gaspar de Matos, M. (2019). Positive youth development, thriving, and social engagement: an analysis of gender differences in Spanish youth. Scandinavian Journal of Psychology, 60, 559–568.
We have replaced our reference 21 by one more accurate:
Kozina, A., Gomez-Baya, D., Gaspar de Matos M., Tome, G. and Wiium, N. (2021) The Association Between the 5Csand Anxiety—Insights from Three Countries: Portugal, Slovenia, and Spain. Frontiers in Psychology. 12:668049, and added:
Manrique-Millones, D.L.; Pineda-Marin, C.P.; Millones-Rivalles, R.B.; Dimitrova, R. The 7Cs of Positive Youth Development in Colombia and Peru: A Promising Model for Reduction of Risky Behaviors Among Youth and Emerging Adults. In Handbook of Positive Youth Development; Dimitrova R., Wiium N., Eds Springer: Cham, Switzerland, 2021; pp. 35–48. https://doi.org/10.1007/978-3-030-70262-5_3
- Are the manuscript’s results reproducible based on the details given in the methods section?
Yes
Reply: Thank you for your comment.
- Are the figures/tables/images/schemes appropriate? Do they properly show the data? Are they easy to interpret and understand? Is the data interpreted appropriately and consistently throughout the manuscript? Please include details regarding the statistical analysis or data acquired from specific databases.
Yes
Reply: Thank you for your comment.
- Are the conclusions consistent with the evidence and arguments presented?
Yes
Reply: Thank you for your comment.
- Please evaluate the ethics statements and data availability statements to ensure they are adequate.
Ethics statement and data availability statement are adequate.
Reply: Thank you for your comment.
Specific comments
- Line 13: I believe you are referring to depression as a problematic behavior, not a problem behavior.
Reply: In our revision, we have instead used the term “adjustment problems” throughout the manuscript, which we believe is more accurate.
- Please carefully check the writing of the abstract. There are some grammatical mistakes that should be taken care of. It is difficult to read, especially because of the absence of commas.
Reply: Thank you. We have looked at the Abstract and addressed the grammatical mistakes.
- English should be revised throughout the manuscript, since there are several grammatical, verbal and spelling mistakes.
Reply: We have now carefully read the manuscript and revised the English.
- In the first paragraph of the introduction (lines 31-39), the five Cs of Lerner et al. are explained. However, there is no information about the sixth C, contribution, which is the main goal of this specific PYD framework. Contribution arises when the other five Cs are present, I think there should be at least a mention about it.
Reply: We have now added some information about contribution, as the sixth C in the introduction.
Modification in the manuscript page 1 line 39-41: Lerner proposes that when these 5Cs are present in a young person, a sixth C, contribution, emerges, where the young person can constructively contribute to self, family, community, even civil society (Lerner, 2004).
- Line 40, you say PYD for referring to positive youth development. However, previously you do not mention anything about PYD meaning. Please, include this acronym between brackets the first time the term positive youth development is used in the main text.
Reply: Thank you for this comment. The acronym PYD for Positive Youth Development has now been added in line 31, when it was first mentioned.
- Line 46, there should be stated “problematic behaviors” instead of “problem behaviours”. Please use the same style of English, either American or British (behavior/behaviour).
Reply: In our revision, we have replaced the term “problem behaviours” with “adjustment problems” and also used American English throughout the manuscript.
- I am not really sure whether depression is a problematic behavior like violence, delinquency or drugs/alcohol use. Why do you consider it as a problematic behavior instead of an illness?
Reply: We have now replaced problem behavior with adjustment problems.
- Please check and correct throughout the whole manuscript the term “problem behavior”.
Reply: Thank you, we have followed the advice and replace problem behavior with “adjustment problems”, which we believe is a more accurate term.
- Some connectors do not make sense with the information of the text (e.g. line 81).
Reply: We have followed the reviewer’s advice and rephrase the text.
Modification in the manuscript page 2 line 82-84: “In line with Geldhof et al. [14], the positive association between Caring and adjustment problems (i.e., depressive symptoms) can be explained by emotional sensitivity or a tendency for excessive concern about the feelings and thoughts of others.”
- L110: Prominent. Do you mean predominant?
Reply: Yes, thank you. We meant “predominant” and have now made the correction.
- Why do you believe this research is important for PYD literature and depression field of study? This reflection should be included in section 1.4 (L157). Reading the introduction and theoretical framework, I still do not clearly see why is this research important, what is the differential element that makes it useful for the field of study. There are plenty of references addressing PYD (and specifically the 5Cs model) and depression. What is different with your study?
Reply: Thank you for your comment. We have added information regarding the justification of the study in the section on “the present study” and renamed it as “Justification and Objectives of the present study”.
Modification in the manuscript page 4 lines 154-163: “Even though some studies have been done on the association between thriving indicators and depressive symptoms with European samples (e.g., Croatia and Spain), more research, such as the present study is needed in other cultural contexts and especially comparative ones. Moreover, if the protective role of the 5Cs are confirmed, future intervention that promotes these thriving indicators would likewise help young people deal with their mental health also in our Peruvian and Spanish contexts. Furthermore, although Peru and Spain are both Spanish speaking countries, the cultural differences between them could reflect in the findings regarding the role of thriving indicators on depressive symptoms, a topic that warrants more attention, and which ultimately can help tailor interventions to meet specific needs.”
- In the participants section, is there any selection criteria for the sample? In university, there can be people from 18 to 65+ years, but PYD framework addresses only youth people.
Reply: The inclusion criteria was age and concerned only young people aged 18 to 29 years. This has been made clear in the participant’s section.
Modification in the manuscript page 5 lines 196-197: As inclusion criteria, young people aged 18 to 29 years, were selected to participate in the study.
- Checking the validation of the instruments used, I see that the original questionnaire designed in the US (reference 14) was validated for grade 5 to grade 12 students (i.e. up to 18 years, what in Spanish would be “bachillerato”). However, the sample from this study is older than that. Has there been any validation on people up to 29 years? (I believe that is the age limit you used, since it is not clear in the manuscript and is only addressed in the introduction as young adults up to 29 years).
Moreover, when it comes to the Spanish translation of the instrument, I cannot see any validation in the reference 21 regarding the PYD-SF. This reference only shows a cross-sectional study from the 2nd author of the present manuscript in which a translation of PYD-SF has been carried out, but it has not been validated. I also miss the validation in terms of age, as I stated previously.
Reply: The psychometric properties of the 5Cs of PYD scale has been verified in American adolescent samples (Bowers et al., 2010; Geldhof et al., 2014; Phelps et al., 2009) as well as in emerging adults (Dvorsky et al., 2019). Additionally, its validity has been confirmed in many other contexts, for example, samples of young people in Europe, in Ireland (Conway, 2015), in Lithuania (Erentaite & RaižienÄ—, 2015), in Italy, Bulgaria, Rumania and Norway (Dimitrova et al., 2021) and also outside Europe (i.e., New Zealand, Fernandes et al., 2021). In the Spanish context the psychometric properties of the scale have been confirmed in previous studies (Gómez-Baya et al., 2019; Kozina et al., 2021), and also in South America, Perú and Colombia (Manrique-Millones, 2021). We have added these references in the manuscript.
Dvorsky, M. R., Kofler, M. J., Burns, G. L., Luebbe, A. M., Garner, A. A., Jarrett, M. J., et al. (2019). Factor structure and criterion validity of the Five Cs model of positive youth development in a multi-university sample of college students. Journal of Youth and Adolescence, 48(3), 537–553. https://doi.org/10.1007/s10964-018-0938-y
Gomez-Baya, D., Reis, M., and Gaspar de Matos, M. (2019). Positive youth development, thriving, and social engagement: an analysis of gender differences in Spanish youth. Scandinavian Journal of Psychology, 60, 559–568.
We have replaced our reference 21 by one more accurate:
Kozina, A., Gomez-Baya, D., Gaspar de Matos M., Tome, G. and Wiium, N. (2021) The Association Between the 5Csand Anxiety—Insights from Three Countries: Portugal, Slovenia, and Spain. Frontiers in Psychology. 12:668049, and added:
Manrique-Millones, D.L.; Pineda-Marin, C.P.; Millones-Rivalles, R.B.; Dimitrova, R. The 7Cs of Positive Youth Development in Colombia and Peru: A Promising Model for Reduction of Risky Behaviors Among Youth and Emerging Adults. In Handbook of Positive Youth Development; Dimitrova R., Wiium N., Eds Springer: Cham, Switzerland, 2021; pp. 35–48. https://doi.org/10.1007/978-3-030-70262-5_3
- Character’s α value in Spanish sample is low to be considered acceptable.
Reply: Yes, we agree with you and have addressed the low Cronbach’s alpha of Character in the Spanish sample as a limitation in our revision.
Modification in the manuscript page 5 lines 242-246 “…and acceptable coefficients were found in the overall scale (α = .876) and in four Cs (Competence: α = .725; Confidence: α = .771; Connection: α = .770; Caring: α = .822) with the exception of Character (α = .586) that was rather low for the Spanish sample.”
Modification in the manuscript page 13 lines 471-473: “Fifth, reliability coefficient for the PYD dimension, Character in the Spanish sample was rather low, thus results involving this variable should be interpreted with caution.”
- In 3.1 section, the Connection results are missing in the first paragraph (L273-282). The other Cs are all addressed.
Reply: Thank you for this comment. The non-significant result of Connection has now been added in the result section.
Modification in the manuscript page 6 lines 285-286: “No gender difference was found for Connection, the overall PYD score, nor for the score on depressive symptoms.”
Modification in the manuscript page 6 lines 289-290: “No country differences were found for depressive symptoms, Connection or the overall PYD score.”
- L343 – Change the square brackets for parentheses.
Reply: Thank you. This has now been corrected.
- L416 – Please change “preoccupied with” for “worried about.”
Reply: The word “preoccupied” has now been replaced with “worried about”.
- L420 – Indent in first line is missing.
Reply: Indentation has now been used in line 420.
- It could be interesting for your further research to address more than one theoretical model of PYD. There are several PYD frameworks that could help you to gain insight into the PYD-depression field, such as the Developmental Assets Profile (Benson, 1997), the Domains of Learning Experiences (Dworkin et al., 2003; Larson et al., 2006), or the more recent Critical PYD (Gonzalez et al., 2020).
In Spain, Oliva et al. (2010) designed an educational PYD model through teamwork among psychologists, psychiatrists and education professionals (Oliva, A., Ríos, M., Antolín, L., Parra, Á., Hernando, Á., & Pertegal, M.-Á. (2010). Más allá del déficit: construyendo un modelo de desarrollo positivo adolescente. Infancia y Aprendizaje, 33(2), 223-234. https://doi.org/10.1174/021037010791114562).
I think the work from Olenik (2019) could also help you to dig deeper and gain more insight about this topic (Olenik, C. (2019). The Evolution of Positive Youth Development as a Key International Development Approach. Global Social Welfare, 6(1), 5-15. https://doi.org/10.1007/s40609-018-0120-1)
Reply: Thank you very much for the suggested references. We have used the Olenik reference and plan to incorporate information on other PYD frameworks in a future manuscript.
- L501 – please write correctly the Data Availability Statement
Reply: Thank you, the Data Availability Statement has now been revised.
Modification in the manuscript page 15 lines 539-540: “The data presented in this study are available on request from the corresponding author.”

Reviewer 2 Report
The paper shows a potentially relevant goal, that is, the cross-sectional study of the Positive Youth Development (5C factors) and the relationship with depression symptoms (PHQ-9). Unfortunately, there are very relevant theoretical issue that limit the scope of the study.
Please see the attached file.

Reviewer 3 Report
I appreciate you allowing me to review it. The strengths of this paper are obvious. The variables are very simple and could be understood according to a general developmental model of the subject. Thus, this research topic has the potential to contribute not only to this field but also to the field of mental health more broadly. However, I could not make sense of how your analytical model was objectively derived from previous knowledge. For this, you need to be more specific about the conceptual and theoretical organization among the measurement variables. In particular, I was concerned about the inconsistencies between the analytical model and theories/practical methods.
Major comments:
General: aims
There were several objectives, so it was difficult to know which one to focus on. if the focus is on non-US-contexts, the comparison with existing US-contexts needs to be better described in the discussion. If you intended to examine the 5Cs’ mediating effects, the text should be structured primarily around findings that adequately support that hypothesis. Please note that each paragraph in the Introduction and Discussion was structured by something completely different.
Introduction: Line 41
"Dynamic interaction" was unclear. Please explain what kind of measurement or modeling is used to show this, and specifically what kind of interaction there is.
Introduction: Paragraph #2
I would like to know more specifically about the relationship between PYD and mental health. For example, does PYD and improved mental health (partially) overlap? Or can you explain the causality between the two? Like the comment above, without this information, I cannot determine if the authors' study's hypothesis and interpretation of the data are scientifically derived.
Introduction: section 1.1
Depression as a mental disorder was mixed with depressive symptoms in the healthy sample similar to this study. Please clarify each and mention their relevance to this study. Please review this carefully in the Discussion section as well.
Introduction: section 1.2
Gender differences with respect to PYD and 5Cs were mentioned, but the relationships were unclear, as was the proposed analytical model. Also, no evidence was presented to suggest "mediation".
Introduction & Discussion:
You mentioned the COVID-19 pandemic. How was this affected in this study? There is no comparison to pre-pandemic data, so this impact needs to be discussed carefully. If you are presenting this information in the introduction, please focus on information relevant to the current hypothesis. Conversely, this topic in the Discussion section was arbitrarily limited in scope. Why explain only the high PHQ9 scores when the pandemic should have affected the entire variables and relationships?
Introduction: section 1.3
The social backgrounds related to youth mental health in Peru and Spain were clearly described. Why were the factors indicated in these previous studies not included in the analytical model? For example, "social inequity" and "confinement" were associated with the frequency of depression among young people. In this context, is the model proposed by the authors still valid?
Methods: Participants and Procedure
Inclusion and exclusion criteria for participants were not clearly stated. Please clarify the attributes of the population that this study will cover.
Methods: Participants and Procedure
Does the grade level of the participants need to be controlled for PYD? For example, please test for differences in 5Cs scores and impact on depressive symptoms between first and near-graduation grades. Sensitivity analysis in the current model may also be a useful.
Data Analysis: multiple partial mediation
A multilevel linear mixed model might suit better. Could you build a base model that the 5Cs factors affect depressive symptoms (fixed effects), and that the degree of impact varies by gender and country ( random effects)?
Data Analysis: multiple partial mediation
Using the current model, how could this "mediation" be assumed? I could not draw this relationship from the information in the introduction. Please present the design process of the model in a logical manner. Were the data measures (especially cross-sectional data) appropriate for this model, and are each PYD factors and depressive symptoms independent? The following literature may be helpful on these issues
: Kline 2015 The Mediation Myth. Basic and Applied Social Psychology, 202-213.
Result & Discussion: Demographics & Descriptive statistics
First, please discuss the characteristics of the Peruvian and Spanish data respectively, including a comparison with the US-context. It is unclear what the differences between Peru and Spain mean in the context of commonly known findings. The authors do not address this issue at all, even though it is mentioned in the Introduction.
Result: Linear regression
Because of the low variance explanatory ratio in the regression model, depressive symptoms are not well explained by the 5Cs and gender/age factors alone (especially in Peru). This result should be considered carefully. Also, a mediation model using only factors that explain only 20% of the variance in depressive symptoms has the same concern. Please do not interpret the results as if the underlying mechanism of depressive symptoms has been demonstrated.
Line378-381
Does this previous study correspond to the discussion in this study? There are many other places where each of the 5Cs factors are associated with a different term. You should explain how each concept is organized.
Line 392.
Since "these two pairs of Cs are often found in tandem" is known, that pair relationship should be incorporated into the analytical model. However, the authors left these independent.
Minor comments:
Line 199
Please insert a space for "in2021".
Table 3 & 4
The "1" on the diagonal is unnecessary (sometimes a hyphen).
Round 2
Reviewer 1 Report
Dear authors,
Thank you for all your efforts in improving the manuscript. Now I believe it is properly fit for publication. I hope my comments helped you to improve the manuscript and your future research within PYD field.
Best regards
Reviewer 3 Report
I appreciate the authors' best efforts to address my concerns.